# Nanoscale membranes that chemically isolate and electronically wire up the abiotic/biotic interface

Jose A. Cornejo[1], Hua Sheng [2], Eran Edri [2,3], Caroline M. Ajo-Franklin [1,2] & Heinz Frei [2]

By electrochemically coupling microbial and abiotic catalysts, bioelectrochemical systems such as microbial electrolysis cells and microbial electrosynthesis systems synthesize energy-rich chemicals from energy-poor precursors with unmatched efficiency. However, to circumvent chemical incompatibilities between the microbial cells and inorganic materials that result in toxicity, corrosion, fouling, and efficiency-degrading cross-reactions between oxidation and reduction environments, bioelectrochemical systems physically separate the microbial and inorganic catalysts by macroscopic distances, thus introducing ohmic losses, rendering these systems impractical at scale. Here we electrochemically couple an inorganic catalyst, a $SnO_2$ anode, with a microbial catalyst, *Shewanella oneidensis*, via a 2-nm-thick silica membrane containing -CN and $-NO_2$ functionalized *p*-oligo(phenylene vinylene) molecular wires. This membrane enables electron flow at $0.51\,\mu A\,cm^{-2}$ from microbial catalysts to the inorganic anode, while blocking small molecule transport. Thus the modular architecture avoids chemical incompatibilities without ohmic losses and introduces an immense design space for scale up of bioelectrochemical systems.

[1] Molecular Foundry Division, Lawrence Berkeley National Laboratory, University of California, Berkeley, CA 94720, USA. [2] Molecular Biophysics and Integrated Bioimaging Division, Lawrence Berkeley National Laboratory, University of California, Berkeley, CA 94720, USA. [3] Present address: Department of Chemical Engineering, Ben-Gurion University of the Negev Be'er Sheva, 8410501 Beersheba, Israel. These authors contributed equally: Jose A. Cornejo, Hua Sheng, Eran Edri. Correspondence and requests for materials should be addressed to C.M.A.-F. (email: cajo-franklin@lbl.gov) or to H.F. (email: hmfrei@lbl.gov)

While bioelectrochemical systems have traditionally focused on electricity or hydrogen production, recently a new generation of bioelectrochemical systems has been developed to synthesize inorganic chemicals, fuels, and pharmaceuticals from wastewater or renewable energy[1–7]. These systems use microbial cells and inorganic materials as separate catalysts for both oxidative and reductive reactions, and these two reactions are coupled to make products of interest at high thermodynamic efficiency. When the microbial catalysts perform oxidative reaction and the inorganic material, e.g., Pt[8] or carbon cloth[9], catalyzes the reductive reaction, these systems are called microbial electrolysis cells[8–12]. Alternatively, in microbial electrosynthesis[13–16], biohybrid[17–19], or artificial photosynthesis[20, 21] systems, the direction of electron flow is reversed so that the inorganic material performs oxidative catalysis and the microbial catalyst carries out reductive reaction (Supplementary Fig. 1). Independent of the direction of electron flow, a recognized challenge in these systems is that the microbial and inorganic catalysts require distinct chemical environments for optimal function[4, 6, 7]. For example, inorganic catalysts generate reactive oxygen species[15, 19] or leach heavy metal ions[19, 20], which kill the microbial catalysts, or microbial catalysts can corrode the inorganic catalyst[5] or generate undesired products, i.e., $CH_4$ in $H_2$, via cross-reactions[3, 6]. To achieve and sustain the chemically distinct environments, the biotic and abiotic catalysts are separated by millimeters to centimeters[15, 16, 22] or by macroscale membranes[9, 10]. This separation leads to crippling ohmic losses on the order of 25% of the cell voltage[20, 21] that impair system scale up[1–3, 7] and device architectures that are incongruent with large-scale manufacturing[23]. Thus new concepts to simultaneously chemically separate, yet electrochemically couple, microbial and inorganic catalysts on the shortest possible length scale are needed to render scale up of bioelectrochemical systems feasible.

Similar to abiotic and microbial catalysts, purely synthetic oxidation and reduction catalysts require separate chemical environments for optimal function. Recently, nanoscale silica membranes containing embedded molecular wires have been introduced as a platform to electrochemically couple these catalysts under separate reaction environments at the smallest possible length scale[24]. The 2 nm-thick amorphous silica layer of these membranes is impermeable to $O_2$ and other small molecules but transmits protons[25]. Embedded in the silica layer, conjugated *oligo*-para(phenylene vinylene) molecules can rapidly transfer electrons at energies dictated by their orbital energetics[26–28] between the catalysts.

While these silica membranes with embedded wires offer the features needed, microorganisms that grow, live, and die offer a uniquely challenging physicochemical environment for chemical separation. Likewise, electron transfer in biological catalysts operates in a different kinetic and energetic regime than in synthetic catalysts.

Here we overcome these challenges and show that specifically tailored versions of these silica membranes can indeed couple inorganic catalysts with microbial catalysts on the shortest possible length scale—nanometers—while separating the incompatible abiotic and microbial environments. This proof-of-concept demonstration provides a platform to dramatically reduce ohmic losses associated with macroscale separation while avoiding chemical incompatibilities. This approach to integrate biotic and abiotic catalysts opens up an immense design space for building macroscale systems, thus providing an opportunity for scalable bioelectrochemical systems.

## Results

**Designing the nanoscale membrane**. To explore this concept for the abiotic/biotic interface, we chose to test whether the bacterial

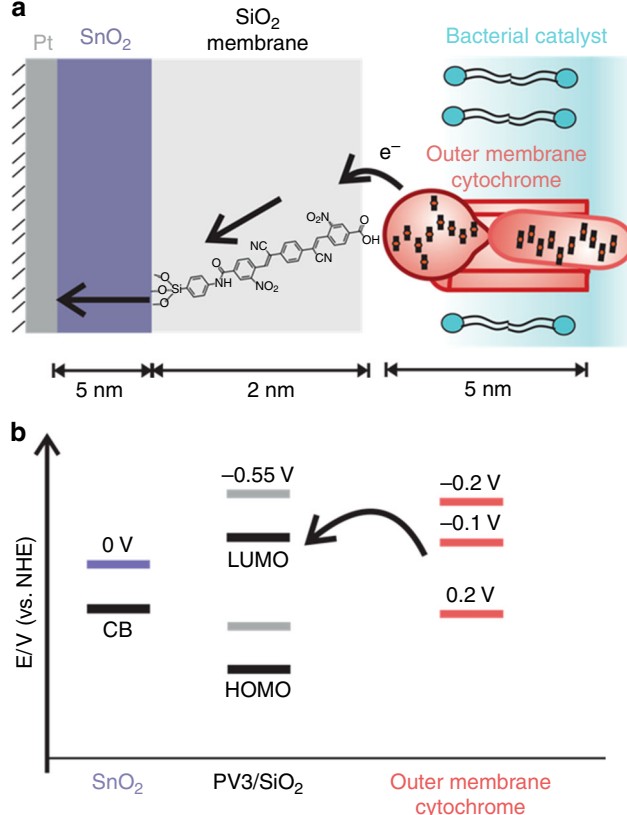

**Fig. 1** Electron transfer path and energy diagram at the designed biotic/abiotic interface. **a** Electrons are transferred from *S. oneidensis* outer membrane cytochrome c to the LUMO of the molecular wire embedded in a 2-nm-thick $SiO_2$ membrane and then to the inorganic catalyst layer ($SnO_2$). **b** Energy levels of outer membrane cytochromes c (red), the PV3 molecular wires (gray), and the $SnO_2$ layer (purple) based on measured values of each component in isolation. When assembled and positively biased in the complete construct, the energy levels of the PV3 wires and $SnO_2$ layer are likely shifted (indicated in black) to more positive potentials

catalyst, *Shewanella oneidensis* MR-1, could be electronically connected and yet chemically isolated from an inorganic catalyst, $SnO_2$, using a nanoscale silica membrane (Fig. 1). *S. oneidensis* MR-1 can oxidize lactate and transfer electrons to a variety of metal oxides via outer membrane cytochromes ($E°' \sim +200$ mV vs. normal hydrogen electrode (NHE); all redox potentials are vs. NHE) directly or in conjugation with flavins ($E°' \sim 0$ mV, $-200$ mV)[29] to provide energy for cell maintenance and growth. We chose $SnO_2$ as the abiotic component because its conduction band is energetically well positioned to be an electron acceptor from *S. oneidensis* (Fig. 1). Since the reduction potential of $-1.7$ V for unsubstituted *oligo*-para(phenylene vinylene) with 3 aryl units[30] is far too negative, we designed an *oligo*-para(phenylene vinylene) molecule (abbrev. PV3) with strong electron withdrawing cyano (CN) and nitro ($NO_2$) substituents whose LUMO (lowest unoccupied molecular orbital) would be better matched to that of the *S. oneidensis* MR-1 outer membrane cytochromes.

**Tailoring electronic properties of molecular wires**. Our first steps were to synthesize this molecular wire and anchor it to the $SnO_2$ anode. The compound 4,4'-((1Z,1'Z)-1,4-phenylenebis(2-cyanoethene-2,1-diyl))bis(3-nitrobenzoic acid) (abbrev. PV3, Fig. 2) was prepared by Knoevenagel condensation strategy

(see Supplementary Methods and Supplementary Figure 2 (NMR)). As shown by cyclic voltammetry (Supplementary Fig. 3), substitution of nitro groups in the terminal aryl rings and cyano groups in the bridging alkene moieties lowers the reduction potential to −0.5 V vs. NHE.

These molecules were covalently anchored on a Pt/SnO$_2$ film in a two-step process. First, the anchoring group 4-(trimethoxysilyl) aniline (abbrev. TMSA) was covalently attached to the oxide surface and its attachment was monitored by infrared spectroscopy and X-ray photoelectron spectroscopy (XPS). The good agreement between polarized Fourier transform infrared (FT-IR) reflection absorption (FT-IRRAS) spectra of free and anchored TMSA (Fig. 3a, traces (1) and (2)) indicates that the silyl aniline

remained intact upon anchoring, as further described in Supplementary Methods. Additionally, a distinct N 1s peak at 399.8 eV (Fig. 3b, trace (2), and Supplementary Fig. 4) appeared in the XPS N 1s spectrum of the SnO$_2$ surface (Fig. 3b, trace (1)) upon TMSA anchoring, providing additional confirmation that TMSA is surface attached[31]. It is worth noting that covalent attachment of the wire to the SnO$_2$ surface via the TMSA anchoring group likely causes electronic effects that determine the exact reduction potential of the wire in the complete assembly.

Next, PV3 was linked to the TMSA anchor via formation of an amide bond (Fig. 1a). Infrared spectroscopy of the resulting PV3 on Pt/SnO$_2$ showed that it shared key spectral features with that of solid PV3 with aniline end groups (Fig. 3a, traces (3) and (4), Supplementary Figures 4 and 6 and spectral assignments in Supplementary Methods). Furthermore, the XPS spectrum of PV3 attached to TMSA on Pt/SnO$_2$ (Fig. 3b, trace (3)) showed a N 1s band centered at 406.2 eV originating from the nitro groups[31] along with overlapping nitrile and amide signals at 400 eV. Taken together, these infrared and XPS analyses confirm that the two-step anchoring method results in the attachment of the intact PV3 wire molecules on the Pt/SnO$_2$ surface.

To complete the nanoscale membrane, atomic layer deposition (ALD) was used to cast the PV3 wires on Pt/SnO$_2$ into SiO$_2$ with a thickness of 1.9 ± 0.3 nm as determined by ellipsometry. While the IRRAS measurement was challenging due to strong

**Fig. 2** Chemical structure of PV3 molecular wire

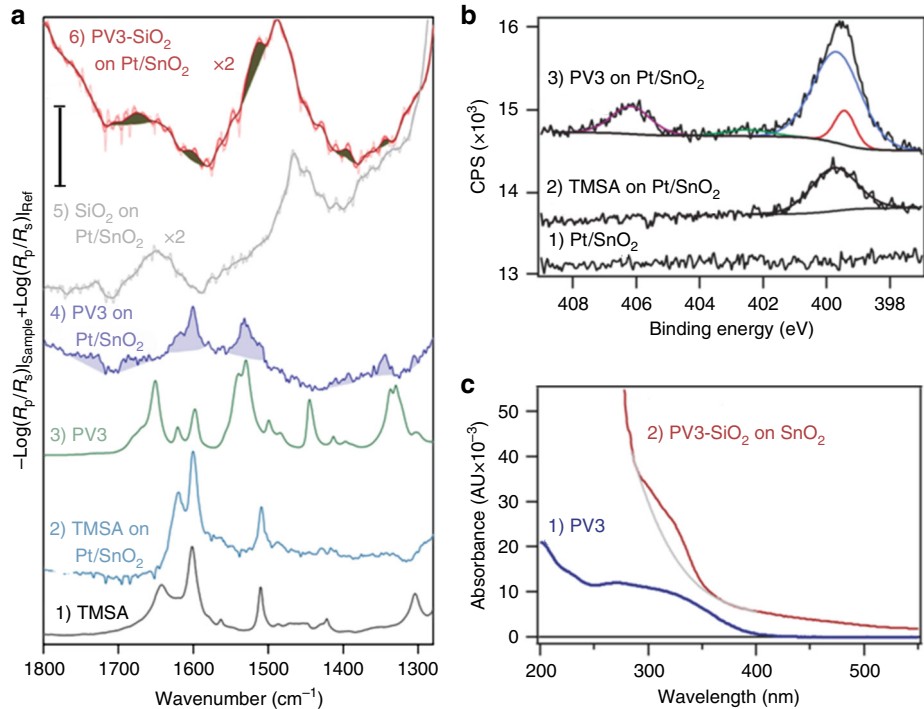

**Fig. 3** Spectroscopic characterization of PV3 wire attachment and casting into SiO$_2$. **a** FT-IR characterization. (1) Absorbance spectrum of TMSA powder in KBr. Scale bar is 0.008. (2) IRRAS of TMSA anchored on Pt/SnO$_2$ (using Pt/SnO$_2$ as reference) at 2 cm$^{-1}$ resolution, computed as the negative log of a single beam spectrum at p-polarization of a sample, divided by a single beam spectrum at s-polarization. Scale bar is 0.002. (3) Absorbance spectrum of powder of PV3 with the aniline groups attached on both ends of the wire molecule (model for TMSA anchored PV3) in KBr. Scale bar is 0.008. (4) IRRAS of PV3 attached to TMSA on Pt/SnO$_2$ (using SnO$_2$/Pt as reference). PV3 bands highlighted in blue. Scale bar is 0.002. (5) IRRAS of pure SiO$_2$ layer on Pt/SnO$_2$ (using Pt/SnO$_2$ as reference). For clarity, raw spectrum shown in light trace is overlaid by computationally filtered (low bandpass) dark trace. Scale bar is 0.002. (6) IRRAS of SiO$_2$-encapsulated PV3 attached to TMSA on Pt/SnO$_2$ (using Pt/SnO$_2$ as reference). PV3 bands are shaded in black for clarity. Scale bar is 0.002. **b** XPS spectrum of the N 1s region of (1) Pt/SnO$_2$; (2) TMSA anchored on Pt/SnO$_2$ showing N 1s NH$_2$ peak; and (3) PV3 attached to TMSA on Pt/SnO$_2$ showing overlapping amide and cyano group signals centered at 399.5 eV and the nitro group at 406.2 eV. Two-component deconvolution is shown (see text). Binding energies are aligned with reference to adventitious C 1s peak at 284.8 eV. CPS counts per second. For completeness, bands for C 1s, O 1s, and Si 2p spectra are shown in Supplementary Fig. 4. **c** UV-Vis spectra of (1) 0.47 μM PV3 in aqueous solution and (2) difference spectrum of SiO$_2$-encapsulated PV3 on quartz/SnO$_2$ (transmission mode). Reference is pure SiO$_2$ on quartz/SnO$_2$. The gray line is a cubic fit for the background

perturbation of the baseline caused by the silica layer (Fig. 3a, trace (5)), we detected four PV3 bands in the PV3-SiO$_2$ on SnO$_2$/Pt (trace (6)). Also, optical spectroscopy revealed an absorption band corresponding to PV3 with a maximum at 323 nm (Fig. 3c and Supplementary Fig. 7) whose intensity indicates a surface concentration of 2 molecules nm$^{-2}$ (see Methods). These observations establish that PV3 wires are intact and dense in the nanoscale silica membrane.

**Chemically isolating two compartments on the nanoscale**. To probe the ability of the PV3-SiO$_2$ on Pt/SnO$_2$ electrode (correct wire electrode) to both chemically separate and electrochemically

connect the SnO$_2$ and *S. oneidensis* catalysts, we needed to establish the level of chemical cross-talk in the absence of the SiO$_2$ membrane and the level of current flow in the absence of embedded wires. Thus we synthesized Pt/SnO$_2$ electrodes (bare electrode) as a negative control that should be unable to provide chemical separation and Pt/SnO$_2$ overlaid with 2 nm SiO$_2$ electrodes (no wire electrode) that should be unable to provide electrochemical coupling.

To probe whether the membrane could chemically separate yet protonically connect the SnO$_2$ catalyst from the aqueous compartment that will contain *S. oneidensis*, we performed cyclic voltammetry in bioelectrochemical reactors (Fig. 4a inset) containing 1 mM K$_3$Fe$^{II}$(CN)$_6$ and K$_4$Fe$^{III}$(CN)$_6$ solution in the aqueous compartment and bare, no wire, or the correct wire electrodes. The ferricyanide couple should be able to reduce and oxidize any accessible SnO$_2$ surface but should be unable to reduce or oxidize the PV3 molecule in the no wire or correct wire electrodes. As expected for the bare electrode, a redox wave is observed at 314 mV (Fig. 4a, inset), confirming that ferricyanide can contact the electrode surface. By contrast, cyclic voltammograms of both the no wire and correct wire electrodes show no significant redox signal at 314 mV (Fig. 4a). (Note that in Fig. 4a the $y$ axis of experiments with SiO$_2$ membrane spans 0.25 mA cm$^{-2}$, whereas the $y$ axis of experiments with bare SnO$_2$ (inset) spans 8 mA cm$^{-2}$.) The absence of a redox wave with the no wire and correct wire samples indicates that the SnO$_2$ surface is >99% inaccessible, demonstrating that the 2-nm-thick SiO$_2$ membrane chemically separates the SnO$_2$ catalyst from the aqueous compartment. It is also essential that this membrane allows H$^+$ transport to enable full electrochemical coupling. The cathodic peak at −0.4 V in both the no wire and correct wire samples, which arises from the reaction of H$^+$ with Pt to form H-Pt, also demonstrates that H$^+$ can move through the silica layer in accordance with past observations[25]. Thus we conclude that the casting of SnO$_2$-anchored wires into a 2 nm silica layer creates a nanoscale membrane that can transmit H$^+$ between the inorganic catalyst and aqueous compartment, while blocking small molecule transport.

**Electron transport across the membrane with wires**. We next sought to determine whether the SiO$_2$ membranes with embedded wires could enable electron flow from the microbial catalyst to the SnO$_2$ catalyst. To do so, we monitored current flow from *S. oneidensis* MR-1-expressing green fluorescence protein (GFP) to the bare, no wire, and correct wire electrodes in microaerobic

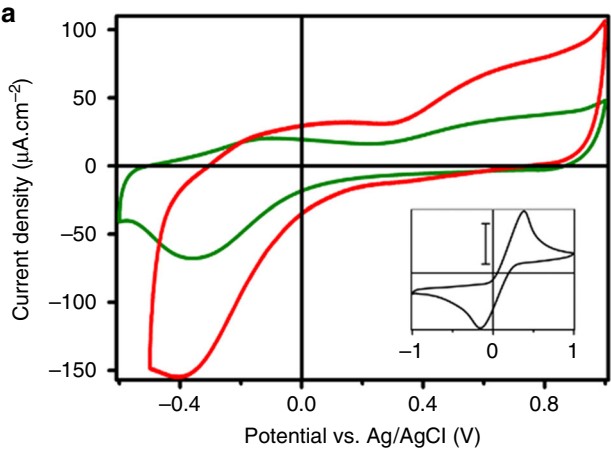

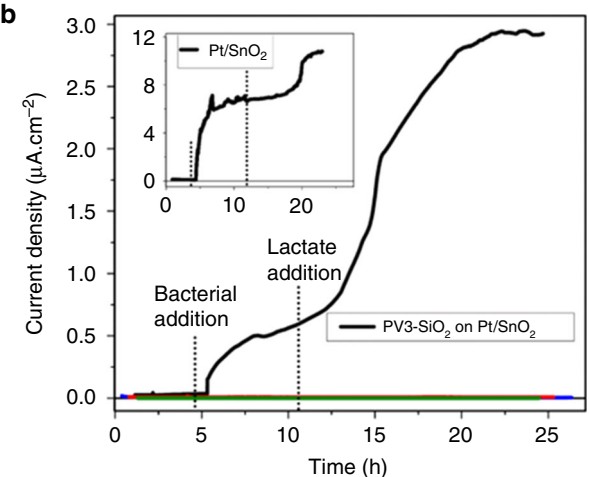

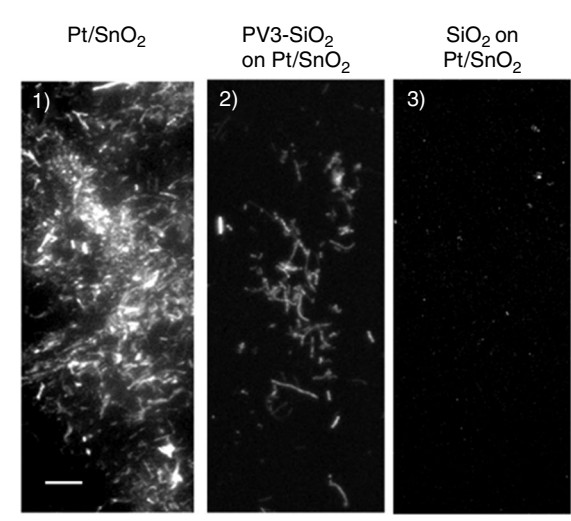

**Fig. 4** Chemical, protonic, and electron transport across membrane. **a** Cyclic voltammograms of pure SiO$_2$ on Pt/SnO$_2$ electrode (no wires, red trace) and SiO$_2$-encapsulated PV3 on Pt/SnO$_2$ electrode (correct wires, green trace). Inset: Redox wave of Fe$^{II}$(CN)$_6$/ Fe$^{III}$(CN)$_6$ with Pt/SnO$_2$ electrode (bare anode). Scale bar 2.5 mA cm$^{-2}$. These scans were performed at a rate of 0.1 V s$^{-1}$ in the presence of 1 mM of K$_4$Fe$^{II}$(CN)$_6$ and K$_3$Fe$^{III}$(CN)$_6$ aqueous solution. The cathodic peak at −0.4 V arises from the reaction of H$^+$ with Pt to form H-Pt. **b** Chronoamperometry at +0.6 V vs. of *S. oneidensis* on SiO$_2$-encapsulated PV3 on Pt/SnO$_2$ electrode (correct wires, black trace); pure SiO$_2$ on Pt/SnO$_2$ electrode (no wires, blue trace); SiO$_2$-encapsulated PV3_SO$_3^-$ on Pt/SnO$_2$ electrode (wrong wires, red trace) and *S. oneidensis* Δ*mtrB* mutant with SiO$_2$-encapsulated PV3 on Pt/SnO$_2$ electrode (green trace). Onset of current coincides with addition of *S. oneidensis* to the electrochemical cell after a period of current stabilization. Second rise of current indicates addition of lactate. Inset: *S. oneidensis* on Pt/SnO$_2$ (bare anode). **c** Confocal fluorescence microscopic images measured after 24 h chronoamperometric measurement presented in **b** of Pt/SnO$_2$ electrode (1); PV3-SiO$_2$ on Pt/SnO$_2$ electrode (2); and pure SiO$_2$ on Pt/SnO$_2$ electrode (3). Scale bar is 10 μm for all images

**Table 1 Current density and bacterial cell density**

| | Sample type | $I_s - I_0$ μA cm$^{-2}$ | Initial OD at 600 nm | Final OD at 600 nm | Cell density cells, μm$^{-2}$ |
|---|---|---|---|---|---|
| Bare anode | Pt/SnO$_2$ | 4.92 ± 2.98 | 0.15 ± 0.00 | 0.24 ± 0.01 | N.D. |
| No wires | SiO$_2$ on Pt/SnO$_2$ | −0.01 ± 0.01 | 0.18 ± 0.05 | 0.07 ± 0.01 | 0.14 ± 0.18 |
| Correct wires | PV3-SiO$_2$ on Pt/SnO$_2$ | 0.51 ± 0.42 | 0.14 ± 0.02 | 0.12 ± 0.01 | 0.41 ± 0.11 |
| Wrong wires | PV3_SO$_3^-$ SiO$_2$ on Pt/SnO$_2$ | −0.06 ± 0.09 | N.D. | N.D. | N.D. |

Data for different electrode and membrane combinations are shown

bioelectrochemical reactors (Supplementary Fig. 8). To ensure that the current we measured was limited by charge transport from *S. oneidensis* to the SnO$_2$[32], we added lactate (the electron donor for *S. oneidensis*) in excess, included Pt wire as a cathode, and poised all SnO$_2$ anodes to a potential of +600 mV vs. NHE (Supplementary Fig. 9). It is important to note that the use of the Pt/SnO$_2$ as an anode and an applied bias is only used to confirm current flow in these proof-of-concept experiments; in a bioelectrochemical system, the Pt layer would be replaced by an inorganic catalyst. The chronoamperometric curves (Fig. 4b) present 28 h of bacterial current generation where additional lactate was introduced toward the end of the experiment to maintain the viability of the bacterial cells. To adjust for sample-to-sample variation in background (abiotic) current density, we calculated the difference between the maximal current produced in the presence of the bacterial catalyst prior to adding fresh lactate ($I_s$) and the current before addition of the bacterial catalyst ($I_0$). All current densities reported below are this $I_s - I_0$ value.

As expected, the reactors containing *S. oneidensis* and bare electrodes produced an average current density of 4.93 ± 2.98 μA cm$^{-2}$ (*n* = 5, Fig. 4b inset, Table 1). This high current density is consistent with the high surface contact between the redox active SnO$_2$ and the bacterial outer membrane cytochromes[29, 33]; however, these electrodes provide no chemical separation between these catalysts (Fig. 4a). For the no wire anodes, no bacterial current was detected above a noise level of 0.01 μA cm$^{-2}$ (*n* = 4, Fig. 4b, Table 1). By contrast, bacteria on correct wire electrode achieved an average current density of 0.51 ± 0.42 μA cm$^{-2}$ (*n* = 4, Fig. 4b, Table 1), which was at least by a factor of 10 above, and on average 51 times greater than, the background current noise in the no wire electrodes. This significant increase in current when the embedded wires are present demonstrates that they transport electrons from the biotic compartment to the inorganic SnO$_2$ layer.

To confirm that these electrons originated from microbial catalyst, we introduced additional electron donor, lactate, to the reactors after 15 h, which should increase the current produced by *S. oneidensis*. In the reactors containing no wire anodes, the average current density did not change significantly ($\Delta I = -0.04 \pm 0.04$ μA cm$^{-2}$), confirming that there is no significant electron transport across the silica. In marked contrast, the current density in reactors with the correct wire electrodes increased on average by 0.19 ± 0.13 μA cm$^{-2}$, or ~35%. This increase in current upon lactate addition confirms that the current in the embedded wire samples originates from oxidation of lactate by the microbial catalysts.

To further confirm that the electron transport occurred through the outer membrane cytochrome *c* and the embedded PV3 wires, we probed the electrochemical behavior of systems in which the energetics of the embedded wire or the bacterial catalyst are unfavorable for electron transfer. In one case, we probed the electrochemical behavior of wild-type *S. oneidensis* with SiO$_2$ membranes containing PV3 molecular wires featuring just a sulfonate substituent (wrong wire electrode) that are energetically unfavorable acceptors for the bacterial outer membrane cytochromes, i.e., the one-electron reduction potential is 1.5 V more negative than the reduction potential of the outer membrane cytochromes[29, 34]. In the other case, we tested the electrochemical behavior of correct wire electrode using a *S. oneidensis* mutant that cannot carry out extracellular electron transfer because it does not display the key outer membrane cytochromes (*S. oneidensis ΔmtrB*)[35]. Both the wild-type *S. oneidensis* with wrong wire anode and the *S. oneidensis ΔmtrB* mutant with the correct wire anode produced only background-level current, −0.06 ± 0.09 μA cm$^{-2}$ and 0.00 ± 0.00 μA cm$^{-2}$, respectively (Fig. 4b, red and green traces). We conclude that the presence of outer membrane cytochromes and proper matching of the orbital energetics of the embedded wire molecules with the potential of the outer membrane cytochromes are essential for electron transfer to occur.

**Bacterial catalysts are viable in the presence of membrane**. Under the reactor conditions, *S. oneidensis* can use electron transfer to an extracellular electrode to maintain or increase cell mass. To determine whether the rate of electron transfer was sufficient to support maintenance or growth of the bacterial catalyst, we monitored the density of *S. oneidensis* cells in solution via optical density at 600 nm (OD$_{600nm}$) and on the electrode via confocal microscopy. *S. oneidensis* can attach and grow in the presence of different electrode surfaces under aerobic conditions and free of electrical bias (Supplementary Fig. 10), which confirms that the surfaces themselves are not toxic. Under microaerobic, polarized conditions, the initial cell density was similar in all the reactors independent of the electrode surface. After 2 days, the cell density in solution dropped by ~60% in the no wire electrodes, yet only decreased ~15% in the reactors containing correct wire electrodes (Table 1). In agreement with this trend, a significant density of bacterial cells (0.41 ± 0.11 cells μm$^{-2}$) were attached to the correct wire electrodes, while a much lower density (0.14 ± 0.18 cells μm$^{-2}$) were attached to the no wire electrode (Fig. 4c). These data indicate that the PV3 wires support electron transfer at a rate that enables *S. oneidensis* to maintain biomass, thus fulfilling an important prerequisite for any bioelectrochemical system.

## Discussion

While providing proof-of-concept of a nanoscale separation membrane for bioelectrochemical systems, the current density from the bacterial catalysts to the molecular wires is presently ~10% of the current density of the bacterial catalysts on the bare Pt/SnO$_2$ electrode. Thus additional understanding and optimization is required to fully realize the efficiency and scalability of this platform. We hypothesize that optimizing the wire redox potential and density will increase the current density from the bacterial catalysts to the inorganic surface via silica-embedded wires so that the rate and energy efficiency of this system will be limited solely by the catalytic components rather than charge transport between them. Specifically, for shifting the LUMO energetics of the wire molecules to more positive values, aryl moieties can be modified by CF$_3$ groups as previously reported for organic molecular wires[36].

While the planar configuration of the platform allowed proof-of-concept, bioelectrochemical systems featuring ultrathin separation membranes require the development of three-dimensional (3D) geometries for extending the separation across all length scales from nano to macro. At the same time, the high surface area of 3D systems will compensate for the relatively slow rate of heterogeneous electron transfer between an electrode and *S. oneidensis* or other microbial catalysts (~100 fA per cell)[37]. We envision core–shell microtube arrays where the core of each tube is the inorganic catalyst, while the shell is an ultrathin silica tube with embedded wires. Such core–shell tube arrays of ~5 cm$^2$ size are being developed in our laboratory[24, 38] along with metal nanocatalyst attachment on the SnO$_2$ layer for important reduction reactions such as the generation of hydrogen peroxide from O$_2$ or conversion of nitrobenzene to aniline.

More broadly, the nanoscale membrane presented here is one specific implementation of a broader concept for improving scaling up in bioelectrochemical systems. In electrosynthesis systems[13–15] and related bioelectrochemical systems[19, 20], microbial catalysts can accept electrons from an electrode at redox potentials ranging from −300 mV to +310 mV[39–43] and utilize these electrons to synthesize biomass[39] or precursors to energy-rich molecules[17]. Since molecular wires with LUMOs spanning −0.5 to −1.7 V or HOMOs (highest occupied molecular orbitals) from +1.4 to over 2 V have already been demonstrated, a nanoscale membrane with the appropriate wires could be used as an interstitial layer between a water-splitting anode and the microbial catalysts, obviating the need for external wires (Supplementary Fig. 9). Electrons and protons from the water-splitting reaction would transit the silica membrane to reach the microbial catalysts, while toxic reactive oxygen species and metal ions would be blocked. Since the electrochemical cycle is closed on the nanoscale, ohmic losses between the water splitting and microbial catalyst would be ~10 mV compared the ~250 mV found in the state of the art. We also envision the nanoscale membrane described here could be used as current collectors[44, 45] or separators[46] to avoid corrosion or oxygen cross-over in microbial fuel cells or microbial electrolysis cells (Supplementary Fig. 1; note that the ultrathin silica membrane is O$_2$ impermeable). Additionally, the concept demonstrated here should apply to bioelectrocatalytic systems where a protein catalyst[47–49] is used in place of a microbial catalyst but suffers the same challenges of chemical compatibility and resistive losses. Thus we envision that the modularity inherent in this membrane architecture will enable tailoring of membranes for a variety of microbial catalysts that provide energy to an inorganic catalyst or vice versa.

In conclusion, we demonstrate a concept for completing the redox couple between a microbial catalyst and inorganic catalyst at the nanoscale while separating the incompatible anodic and cathodic reaction environments. This electron transport occurs only when the energetics of the microbial catalyst and molecular wires are matched, and it occurs rapidly enough to allow the microbial catalyst to maintain biomass. The ability to optimize this platform for different combinations of inorganic and microbial catalysts will drive development of scalable bioelectrochemical systems that harness the energy found in biomass or renewable sources to a variety of chemicals and materials.

## Methods

**Pt deposition**. Pt 100 nm (99.99%) was deposited by e-Beam evaporation (Semicore SC600 e-beam evaporator) at $<2 \times 10^{-6}$ Torr on a Si wafer with 3–5 nm of Ti or Cr to improve adhesion.

**SnO$_2$ deposition**. ALD of tin dioxide was carried out using an Oxford FlexAl-Plasma Enhanced Atomic Layer Deposition system. At a temperature of 200 °C and 80 mTorr (60 SCCM O$_2$; 100 SCCM Ar; SCCM = standard cubic centimeters per minute), Sn precursor (tetrakis(dimethylamido)tin(IV) bubbled with 100 SCCM of Ar when pulsed, 20 SCCM otherwise) was pulsed to the chamber (0.5 s) followed by purging 5 s with 60 SCCM O$_2$ and 200 SCCM Ar. A 0.5 s pre-plasma step with pressure set at 15 mTorr (60 SCCM O$_2$; 100 SCCM Ar) was followed by a 5 s step of 300 W plasma (60 SCCM O$_2$; 20 SCCM Ar) and 1 s step of post-plasma purge with (60 SCCM O$_2$; 100 SCCM Ar).

**TMSA attachment to SnO$_2$**. Two $6.35 \times 1.1$ cm$^2$ substrates (either Si/Pt/SnO$_2$ or fused quartz/SnO$_2$) were cleaned by sonication in isopropanol for 5 min before being arranged back to back in a 25 mL Schlenk flask containing 5.4 mg (0.025 mmol) TMSA (90%; Gellest). The flask was sealed and evacuated for 1 h at ambient temperature before 25 mL of toluene (HPLC grade; Sigma Aldrich) was added by syringe. The content was sonicated for 5 min in a small sonication bath, and the solution refluxed for 12 h. After cooling, the substrates were transferred directly to fresh toluene and sonicated for 10 min, and the solvent was switched to a 50% vol. methanol–toluene mixture and sonicated for 10 min again before changing the solvent to methanol and sonicating for another 10 min. As a final step, the substrates were dried with a N$_2$ stream and immediately utilized for PV3 attachment or kept in a closed vial for further characterization.

**PV3 attachment to TMSA**. Two substrates with TMSA attached (either Si/Pt/SnO$_2$ or quartz/SnO$_2$) were placed back to back in a 10 mL Schlenk flask such that the samples were vertical inside the flask. After addition of 3.1 mg (6.1 μmol) of PV3_CN_NO$_2$_CO$_2$H (4,4′-((1Z,1′Z)-1,4-phenylenebis(2-cyanoethene-2,1-diyl)) bis(3-nitrobenzoic acid)) and 5.4 mg (14 μmol) of HBTU (N,N,N′,N′-tetramethyl-O-(1H-benzotriazol-1-yl)uronium hexafluorophosphate; Sigma Aldrich), the flask was sealed and evacuated on the Schlenk line for 2 h before 10 mL dimethyl formamide (DMF; anhydrous; Sigma Aldrich) was added by syringe with 0.1 mL (0.57 mmol) N,N-diisopropylethylamine (Alfa Aesar). The solution was stirred at 40 °C for 12 h. To clean the substrates, they were rinsed with water and transferred to a flask containing deionized water, sonicated for 5 min, and dried with N$_2$. The cleaned substrates were transferred to the ALD chamber or sealed and kept in the dark for further characterization.

**SiO$_2$ atomic layer deposition**. SiO$_2$ was deposited in a modified Savannah 100 Cambridge Nanotech ALD system equipped with a hollow cathode plasma source and a grounding grid above the sample. SiO$_2$ deposition was carried out at 80 °C with 250 W plasma power using the following cycle: Under a flow of 5 SCCM of Ar, the chamber exhaust was closed and a 0.05 s pulse of tris-dimethylaminosilane (99+% STREM chemicals; ambient temperature) was introduced to the chamber. The chamber was kept sealed for 60 s before purging with 40 SCCM of Ar for 30 s followed by a 5 SCCM 45 s purge with O$_2$. At a pressure of ~200 mTorr, the plasma was ignited for 30 s followed by purging with Ar for 45 s at 5 SCCM. Twenty such cycles lead to a deposition of $2.28 \pm 0.27$ nm-thick SiO$_2$ layer.

**IRRAS measurements**. IRRAS spectra were measured with a Bruker FT-IR spectrometer model Vertex 70 equipped with a LN$_2$ cooled HgCdTe detector, a reflection accessory Bruker model A513/Q, and wire-grid polarizer model F350. The mirror angle of the IRRAS accessory was fixed at 80° and the grid polarizers were switched between an s- and p-polarization (determined by scanning for maximum and minimum infrared throughput (ADC count) for each polarization). Ten spectra of 256 scans each were recorded and averaged to reduce the noise level after purging the sample compartment with N$_2$ for 10–20 min. Sample single channel spectrum at p-polarization was divided by a single channel spectrum of the reference sample at the same polarization and the negative logarithm calculated. A similar spectrum of the s-polarization was calculated as well and the difference is shown in Fig. 3. Supplementary Figure 6 shows the single polarization spectrum of each sample.

Spectra of residual atmospheric water vapor in the sample compartment were obtained by recording two sequential $10 \times 256$ scans using Pt mirror, the first after 10 min purging and the second after >30 min. The water vapor spectrum was used to eliminate residual water vapor infrared bands in the sample and reference spectra by spectral subtraction. All spectra were measured at a 2 cm$^{-1}$ resolution using 6 mm aperture and scan velocity of 40 kHz.

**XPS measurements**. XPS was performed with Kratos Axis Ultra DLD system using a monochromatic Al K$_\alpha$ source ($h\nu = 1486.6$ eV), operated at 225 W at a takeoff angle of 0° relative to the surface normal, and pass energy for narrow scan spectra of 20 eV. Spectral analysis was conducted with the software CasaXPS V2.3. Binding energy scale was corrected to adventitious C1s at 284.8 eV.

**Ultraviolet–visible (UV-Vis) measurements**. Fused quartz substrates were used for SnO$_2$ deposition. Procedures for TMSA anchoring and subsequent attachment of PV3_CN_NO$_2$_CO$_2$H molecules were the same as for Si/Pt substrates. UV-Vis measurements were taken with a Shimadzu spectrophotometer model 2450 using an integrating sphere accessory model ISR-2200. The beam is focused on a $2 \times 2$ mm$^2$ area of the sample. A 5 mm slit and slow scan was used to collect the spectra

without any sample in the reference beam path. Difference spectra were calculated by taking the difference of a spectrum of Quartz/SnO$_2$/PV3_CN_NO$_2$_CO_TMSA, with or without SiO$_2$ (where specified), and the same substrate without PV3_CN_NO$_2$_CO$_2$_TMSA. Prior to recording of spectra, the back side of the substrate was cleaned by UV-ozone treatment for 5 min.

Absorption spectra of PV3_CN_NO$_2$_CO$_2$H in solution were collected in transmission. PV3_CN_NO$_2$_CO$_2$H was dissolved in water with the assistance of a few µL of 1 M tetra-butylammonium hydroxide in methanol (final concentration 0.13 mM). The difference spectrum was calculated by subtracting spectrum of the same solution without PV3_CN_NO$_2$_CO$_2$H.

The surface density of anchored wire molecules was calculated as follows: approximating the baseline in the 260–450 nm region by a third-order polynomial (gray curve of Fig. 3c, trace (2)), the absorbance of quartz/SnO$_2$/PV3/SiO$_2$ at 315 nm is determined as 0.006. Using Beer–Lambert law[50] and a measured extinction coefficient for PV3 of $\varepsilon = 20{,}208$ L mol$^{-1}$ cm$^{-1}$, $A/\varepsilon = $ [PV3]· $d$ is calculated as $2.97 \times 10^{-10}$ mol cm$^{-2}$ or 1.79 molecules nm$^{-2}$.

**Electrochemical measurements**. Cyclic voltammetric measurements were carried out using a CH Instruments model CHI604E potentiostat equipped with a Ag/AgCl as reference electrode (Harvard Apparatus Leak-Free reference electrode, 69–0053; 1 mm diameter; 3.4 M KCl) and Pt wire counter electrode housed in a custom-made photoelectrochemical cell. An aqueous solution of 1 mM of K$_3$Fe$^{II}$(CN)$_6$ and K$_4$Fe$^{III}$(CN)$_6$ was used for testing accessibility of the SnO$_2$ surface.

Redox potentials of PV3 and PV3_SO$_3^-$ molecules were calculated from cyclic voltammetry or linear sweep voltammetry measurements of 1 mM solution of these molecules with 0.1 M tert-butyl ammonium hexafluorophosphate (N(t-Bu)$_4$PF$_6$) and ferrocene (1 mM) in dry DMF purged with Ar or N$_2$. The ferrocene (0/+1) redox potential is 0.45 V vs. saturated calomel electrode (SCE). The LUMO of PV3_CN_NO$_2$_CO$_2$CH$_3$ is situated at −0.55 V vs. NHE, and the HOMO level was calculated from the intersection of optical absorption and fluorescence bands as 3.0 eV vs. NHE. Linear sweep voltammetry was used to determine the HOMO level of PV3_SO$_3^-$ tripod, and a value of 1.35 V vs. NHE was determined, in agreement with literature[30].

**Bacterial strains and growth conditions**. Two strains of the S. oneidensis MR-1 were used in this work. To visualize attachment of the bacteria to surfaces, we used S. oneidensis MR-1-expressing GFP[51]. To probe the involvement of outer membrane cytochromes in electron transfer, we used a mutant S. oneidensis MR-1 that does not express outer membrane cytochrome c (S. oneidensis ΔmtrB)[52]. Bacterial cultures were inoculated from frozen glycerol stocks into 5 mL of Luria-Bertani broth and grown at 30 °C with 225 rpm shaking overnight to early stationary phase. Cultures of S. oneidensis MR-1-expressing GFP were grown with 50 µg kanamycin.

**Bioelectrochemical characterization**. After overnight growth, S. oneidensis MR-1 cultures were washed 2 times and then diluted approximately ten-fold to a final OD$_{600}$ of ~0.15 (~2.5 × 10$^7$ cells mL$^{-1}$) into an electrochemical reactor containing M9 minimal salts (Beckton, Dickinson and Co.), 80 mM sodium lactate, and 30 mM sodium fumarate. To maintain a microaerobic environment, the electrochemical reactor was continuously purged with N$_2$(g).

Chronoamperometry was carried out using a Bio-Logic Science Instruments potentiostat model VSP-300. Single-chamber electrochemical cells used for measurements (Supplementary Fig. 8) consisted of a Ag/AgCl reference electrode and Pt wire counter electrode. Open circuit potentials on the working electrodes without bacteria were recorded until they reached equilibrium, followed by monitoring of bacterial current over a period of 24 h. Microbial cultures were introduced in the system after a period of stabilization in which only background current was observed. This period varied between 2 and 5 h. Since the bioreactors were kept under microaerobic conditions, lactate was consumed by S. oneidensis using both oxygen and the anode as a terminal electron donor. To keep the microbial cultures viable past 24 h, additional sodium lactate was added to the electrochemical cells to maintain a lactate concentration of at least 80 mM.

**Confocal laser scanning microscopy**. Confocal imaging by a Zeiss model LSM 710 instrument with Zen software was performed on all samples following electrochemical measurements to monitor microbial attachment on the working electrodes by emission from GFP-expressed S. oneidensis cultures. Images of 512 × 512 pixel arrays were acquired utilizing a 1.4 NA 63× oil-immersion objective and a sample excitation at 488 nm using an argon ion laser.

**Ellipsometry measurements**. Ellipsometry was carried out with a J.A. Woollam variable angle ellipsometer in the 900–250 nm range at angles in the range 45–70° using 5° steps. A range of angles rather than a single angle was used in order to improve the statistical reliability of the result[53]. A B-Spline model was used to describe the organic layer optical properties and a Cauchy model for the SnO$_2$ layer[53].

**Data availability**. The data that support the findings of this study are available from the corresponding authors upon reasonable request.

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

## Acknowledgements

This work was supported by the Laboratory Directed Research and Development Program of Lawrence Berkeley National Laboratory under U.S. Department of Energy Contract No. DE-AC02-05CH11231. Work at the Molecular Foundry was supported by the Office of Science, Office of Basic Energy Sciences, of the U.S. Department of Energy under Contract No. DE-AC02-05CH11231. A portion of the work (XPS and IRRAS) was performed at the Joint Center for Artificial Photosynthesis, a DOE Energy Innovation Hub, supported through the Office of Science of the U.S. Department of Energy under Award No. DE-SC0004993.

## Author contributions

C.M.A.-F. and H.F. conceptualized the design, guided the experimental design and analysis and edited the manuscript. J.A.C. prepared the bacterial samples, developed together with E.E. the electrochemical cell, and conducted the electrochemical experiments and confocal microscopy. H.S. designed and executed the synthesis and spectroscopic characterization of molecular wires. E.E. prepared and spectroscopically characterized the membranes on inorganic substrates. All authors contributed to manuscript writing.

## Additional information

**Competing interests:** The authors declare no competing interests.

