## [Peer Review File · Nature Communications]

Reviewers' Comments:

Reviewer #1:

Remarks to the Author:

The manuscript entitled "Nanoscale membranes that chemically isolate and electronically wire up the abiotic/biotic interface" describes a method to construct an anode for a microbial fuel cell. The anode is based on the author's previously published method to embed molecular wires in a SiO₂ thin layer. For the bacteria activation, the wire was redesigned with withdrawing groups to alter the electronic properties. The use of isolated enzymes or bacteria to generate electrical power is not new and detailed in previously published papers. As the authors mentioned, electrode fouling or self-oxidation/reduction is indeed an issue that needs to be addressed and the strategy presented is interesting. The electrode characterization conducted nicely, but, unfortunately, the designed system suffers from several issues that limit its novelty and practical use, and therefore, not suitable to high impact journal like Nature communication.

a. Although the authors altered the potential of the molecular wire in ca. 1V, the potential is still too negative to act as an electron acceptor for the Cyt C in the bacteria, which is 0.5V more positive. To solve this issue, the authors biased the electrode to allow the anodic currents, which limits any future use. The authors used very strong withdrawing group in their design, therefore, I'm not sure, if it is possible to improve the system and shift the potential.

b. The novelty of the silica coating and the molecular wire is limited, as it has been shown before in several published papers by the authors.

c. The membranous Cyt C can establish electrical contacting directly with the uncoated electrode, therefore, the use of molecular wire become less attractive. A molecular wire that can anchor to bacteria cyt c will make the system much more attractive and novel and will minimize diffusion limitations. The current design is inefficient.

d. The use of biofuel cells and specifically, bioanode for lactate oxidation has been widely used. The authors should add these systems to the manuscript and compare the different strategies.

e. The *S. oneidensis* was used to construct photoelectrochemical and electrochemical cells. As the authors aiming to improve these MFC, they should mention it in the text and compare the advantages of the new system.

f. In table 1, several numbers "jumped" and should be fixed.

Overall, the suggested system is interesting, but not novel enough to be published in Nature communication. Also, suffers from some basic design problems that make it less attractive.

Reviewer #2:

Remarks to the Author:

I enjoyed very much this interesting manuscript that may prove to be revolutionary in the field of bioelectrochemical systems. For many years researchers have been developing BESs and needed to deal with current collection. The approach presented here avoids current collection issues for those systems where two reactions can be combined that do not require additional energy input (although I could think of some options where this could be dealt with as well, indirectly). For me, the key reason to do this is thus current collection, and not necessarily the occurrence of two incompatible reactions as this is often topical.

On the downside, the manuscript needs to be improved in terms of clarity. First, based on the abstract it is next to impossible to understand what is achieved. Why not call it a bioelectrochemical system, rather than biohybrid system which can be many things? There is unfortunately no data in the abstract, and for the reader it is not clear that anodic oxidation of lactic acid is catalyzed by *Shewanella* leading to the reduction of an electron acceptor on the other side. I would also stress the key advantage of current collection here.

Second, the manuscript is quite elaborate on the construction of the multilayer system, and is subsequently quite brief on the effective experiments. I would like to see electron balances for

consumed lactate, showing consumption coupled to a charge transfer that can be quantified. New lactate is added during an experiment, however it is unclear whether the lactate was used by then? Based on the low current I would not expect so. And, was this experiment repeated? Third, I miss an experiment where the Pt layer is directly used with an electron acceptor, e.g. ferricyanide or oxygen. Unless I am mistaken, I have the impression that in all cases an additional counter electrode (Pt wire) was introduced.

Line 40: in one sentence too many combinations are made, which makes unclear what is possible. Additionally, "biohybrid systems" is a new term in a field that does not need new terms unless justifiable. Why not use an existing technology term?

Line 46: centimeters are never needed in practice
Key is to discuss current collection here as well

Line 105: the

By line 140ish I was wondering whether some of the verification outcomes should be discussed in supplementary information, enabling a stronger focus on the results obtained with *Shewanella*

Line 167: I wondered here how the Pt side of the system was connected to

Line 188 and further: can it be demonstrated that the electrode is not able to oxidize hydrogen? At low partial pressures, the lactic can be oxidized well by the *Shewanella* and the anode can work at quite low potentials still

Line 259: *aeruginosa*

Figure 1: I realize the interest of putting all parts of the system at the same scale, however it would be more evident to the reader what is being achieved if instead of terminal complexes a full bacterium would be drawn here, oxidizing lactate and transferring electrons

Reviewer: Korneel Rabaey

Reviewer #3:

Remarks to the Author:

Decision: Major revision with additional (more convincing data) or sort out the logic behind the theory of present study.

This manuscript describes a study that utilize nanoscale thin film formed by novel material, which was specifically designed to fit the electron transfer from *Shewanella* outer membrane cytochromes (75-78). The approach logically make sense to me though the results did not indicate such (175-182). Compared to bare SiO₂ membrane, the embedding of PV3 wire enhanced the current density as well as the bacteria viability. However, compared to the bare electrode surface, Pt/SnO₂, the coated electrode surface inferior in both current density and bacteria viability. If the bare Pt/SnO₂ electrode surface represented a positive control, the logic needs to be laid out more clearly. For example, in a lot of microbial fuel cell studies, the biocatalyst (the biofilms) was directly grown on the electrode surface without chemical or physical separation. Often time, the biggest ohmic loss was due to the introduction of a membrane in between anode and cathode. Therefore, the logic of present study does not come to me naturally. Why the ohmic loss would be reduced by introducing a membrane between the biocatalyst and anode? It would be beneficial if this part can be extrapolated more clearly. Additionally, the results need to be better organized. There appeared to be more than 3 types of electrodes, Pt/SnO₂, SiO₂ on Pt/SnO₂, PV3-SiO₂ on

Pt/SnO₂, and PV3-SiO₂-SO₃ on Pt/SnO₂, though the authors did not mention the last one. Overall, this approach appears to be promising. I would encourage authors to keep trying with different materials and put biocompatibility into one of the consideration.

24-25 needs to give a better description about such system, for example, many researchers use the term bioelectrochemical system (BES) or microbial electrochemical system (MES). Same for 40-42

Two recent published reviews that describe the connection between biocatalyst and abiotic electrode which might be helpful to discuss present topic. "Li, C., Lesnik, K.L., Liu, H., 2017. Stay connected: Electrical conductivity of microbial aggregates. *Biotechnology Advances* 35(6), 669-680."

"Lovley, D.R., 2016. Happy together: microbial communities that hook up to swap electrons. *The ISME Journal* 11(2), 327-336."

26-28 44-46 & 58-60 Why the separation from millimeters to centimeters would be the central challenge? the reason is not clear enough. In a lot of BES study, the biocatalyst the biofilms was directly grown on the electrode surface without chemical or physical separation. it seems that the biggest ohmic loss is between the biomass and the abiotic materials, in this case the membrane. Introducing one more layer of membrane into the assembly seems to increase the overall ohmic resistance. The benefits of introducing a separation membrane will have to be discussed. Does the membrane has higher affinity with the biofilms of *Shewanella*? or help transfer electrons to electrode? or the separation can help exhaustion of catalyst, etc. The strategy to reduce the contact resistance will have to be discussed more thoroughly.

58-60 In the results of the present study, the bare Pt/SnO₂ electrode seems to possess the higher current density and microbial growth than the SiO₂-PV3 coated Pt/SnO₂ electrode. It is hard to declare the incompatibility of the Pt/SnO₂ electrode.

60-62 the linkage between the advance and the scalable device was not really cleared. How the advance can impact the scalability. The cathodic limitation is also a big part as well as the cost of material, and thus the device that could be benefited from this technique should be mentioned more clearly.

75-78 This section is the key concept of the whole article. I would suggest the authors to discuss/justify this statement more clearly. Why the LUMO of the designed polymer would be more closer to the cytochrome? From the attachment examination, it seems that the polymer has been designed according to the authors' intention but the performance indicate otherwise. Therefore, it is essential to further extrapolate the discussion.

172-174 I_s and I_o should be place closer. The current position is confusing.

175-186 the results were confusing. And reactor containing the bare electrode produce more current than the reactor containing Certain membrane prevent the growth of bacteria. The biocompatibility of this SiO₂-PV3 thin film should be discussed.

From the current density plot it seems that the *Shewanella* was still on the phrase of attaching/biofilm forming and current density could still increase over time. From the CLSM figure, we can see that the surface of electrode was far from occupied. Longer experiment period should be considered.

205-208 The difference would be hard to tell if both current densities were only background level. The current density should not be negative. This provide no further support to the conclusion in line 210-212

237-239 it is hard to tell the link when both current densities were at the background level

244-246 Current data is not supportive enough for the claim of present article.

264-268 The change of inorganic properties of the membrane did not seem to enhance the current production, when compare to bare SnO₂/Pt electrode. Therefore, the claim was not supported by the data.

271-272 The "incompatible environment" of biocatalyst needs to be explained.

633-634 Is that a question for the reviewer?

636-637 Please explain the 45-70° and 5° steps more, or give a citation properly.

The scale of figure 1 should be adjusted according to their actually scale.

Figure 3, what is the background, what is the microbial cells? it seems that the bare electrode has the highest density.

Remarks by Authors in bold

Reviewer #1

The manuscript entitled “Nanoscale membranes that chemically isolate and electronically wire up the abiotic/biotic interface“ describes a method to construct an anode for a microbial fuel cell. The anode is based on the author’s previously published method to embedded molecular wires in a SiO₂ thin layer. For the bacteria activation, the wire was redesigned with withdrawing groups to alter the electronic properties. The use of isolated enzymes or bacteria to generate electrical power is not new and detailed in previously published papers. As the authors mentioned, electrode fouling or self-oxidation/reduction is indeed an issue that needs to be addressed and the strategy presented is interesting. The electrode characterization conducted nicely, but, unfortunately, the designed system suffers from several issues that limit its novelty and practical use, and therefore, not suitable to high impact journal like Nature communication.

Response: As described in the opening paragraph of the text, the work presents a new concept – a nanoscale membrane - to address the widely-recognized need to chemically separate the biotic and abiotic environments in bioelectrochemical systems that synthesize chemicals. This concept applies to a microbial electrolysis cell that synthesizes chemicals from biomass and electricity, but also applies to systems in which inorganic photo- or electro- catalysts provide electrons for microbial catalysts to synthesize chemicals from CO₂ or glycerol (microbial electrosynthesis, microbial electrofermentation, artificial photosynthesis or biohybrid systems). Direct contact between microbial cells and an inorganic surface, while offering robust electrochemical coupling, does not provide chemical separation.

Regardless of the direction of electron flow, the abiotic and biotic catalysts require different chemical environments for optimal function, and this incompatibility is frequently cited as a major challenge in bioelectrochemical systems that synthesize chemicals (Ref 4,6,7). For example, inorganic catalysts generate reactive oxygen species (Ref 15,19) or leach heavy metal ions (Ref 19,20) which kill the microbial catalysts, or microbial catalysts corrode inorganic catalysts (Ref 5) or generate undesired products, i.e. CH₄ in H₂, via cross-reactions (Ref 3,6). Thus, these systems use physical distance or macroscale ion-exchange membranes to separate the oxidative and reductive reactions, and the associated ohmic resistances significantly decrease the energy efficiency and the possibility of scale up (Ref 1-3).

By developing a nanoscale membrane and placing it between the microbial cells and the inorganic anodic surface, our work solves two problems simultaneously: (1) it brings the oxidative and reductive processes together on the shortest possible length scale to minimize ohmic losses and (2) it prevents of degradation of the inorganic electrode by the microbes (e.g. fouling) and degradation of the microbial catalyst by the inorganic electrode (e.g. production of reactive-oxygen species). Thus, our work demonstrates a nanoscale membrane as the first such solution to these broadly acknowledged major challenges in bioelectrochemical systems. While the performance of the membrane is not yet optimized, we describe next steps to improve the current efficiency of the ultrathin membrane (lines 231-238).

In response to the reviewer’s remarks, which are echoed by Reviewer #2 and #3, we have edited the the Abstract (lines 24-37), the first paragraph of the main text (lines 39-54), and the first sentence of the Conclusions (lines 271-273) to clarify the key innovations in our work.

a) Although the authors altered the potential of the molecular wire in ca. 1V, the potential is still too negative to act as an electron acceptor for the Cyt C in the bacteria, which is 0.5V more positive. To solve this issue, the authors biased the electrode to allow the anodic currents,

which limits any future use. The authors used very strong withdrawing group in their design, therefore, I'm not sure, if it is possible to improve the system and shift the potential.

R: The statement that the 600 mV potential was applied to move the wire potential 0.5 V more positive is incorrect. We used a three-electrode system to poise the anode to +600 mV to ensure that the current we measured was limited by electron transfer from *S. oneidensis* to Pt/SnO₂, in accord with best practices (Ref 32). We chose a bias of +600 mV because experiments with the bare Pt/SnO₂ anode gave high current at this bias.

Independent of effects from biasing the electrode, the LUMO energetics of the wire molecule will change when it is attached to an inorganic surface such as SnO₂. This change introduces an uncertainty of a few hundred mV in the molecular wire LUMO level. Thus, the molecular wires prepared for this study are a reasonable starting point for exploring the proposed concept.

In light of the crucial proof-of-concept results shown in Figure 3, the reviewer is most likely correct that the wire LUMO potential needs to be shifted to even more positive values to optimize current density. To achieve this, our next steps include further chemical modification of the wire molecule with strongly electron attracting CF₃ groups at the aryl rings which will further shift the potential to more positive values. Precedents for CF₃ modified PV3 molecules have been reported in the literature (new Ref. 36).

In response to the reviewer's comments, we have added text to the manuscript to clarify why we poised the anode to +600 mV (lines 163-168) and included new data supporting this choice (SI Figure S9). We have also described how to further modify the LUMO by adding CF₃ groups to the PV3 wire (lines 236-238) and added reference citing an example of organic wires modified by CF₃ groups (Ref. 36).

b) The novelty of the silica coating and the molecular wire is limited, as it has been shown before in several published papers by the authors.

R: The novelty of this work is the application of a nanoscale membrane to the abiotic/biotic interface, which had not been tested before, making this approach extremely high risk. At the start of this work, it was unclear if embedded wire molecules could be synthesized to be energetically matched for transporting electrons from exoelectrogens. Additionally, it was unclear whether these embedded molecular wires could approach and electronically couple with cyt c in the cell membrane for electron transfer to be rapid enough to support cell viability. Lastly, it was unclear whether the nanoscale membrane would be robust in the presence of living microorganisms, i.e. whether microbes could be in contact with the nanoscale membranes without degrading the silica layer or the embedded molecular wires.

To highlight the high-risk and novelty of this work, we have added text to the introduction that summarize these challenges (lines 63-69).

c) The membranal Cyt C can establish electrical contact directly with the uncoated electrode, therefore, the use of molecular wire become less attractive. A molecular wire that can anchor to bacteria cyt c will make the system much more attractive and novel and will minimize diffusion limitations. The current design is inefficient.

R: This very important comment shows that the reviewer misunderstood the core concept behind this work. As described in the responses to the general remarks above, the core concept is a new membrane simultaneously capable of 1) electrochemical coupling between oxidative and reductive processes on the nanoscale and 2) chemical separation of the incompatible microbial and inorganic catalysts. Direct electrical contact of membranal cytochrome c with the inorganic surface does not meet the second need to chemically separate the abiotic and microbial environments, e.g. chemicals generated by the inorganic component.

We have added new text to the results and discussion (lines 129-135) to emphasize that the direct attachment does not meet these needs.

d) The use of biofuel cells and specifically, bioanode for lactate oxidation has been widely used. The authors should add these systems to the manuscript and compare the different strategies.

R: The proof-of-concept shown in this manuscript, that a nanoscale membrane can electrochemically connect, yet chemically separate a biotic component and an abiotic component, should apply to the enzymes used in bioelectrocatalytic systems.

Per the reviewer's suggestion, we now include a discussion of how these systems can benefit from a nanoscale membrane (lines 261-268) and appropriate references (Ref 47-49).

e) The *S. oneidensis* was used to construct photoelectrochemical and electrochemical cells. As the authors aiming to improve these MFC, they should mention it in the text and compare the advantages of the new system.

R: As described in the response to the general remark and additional text on p.3 and p.4, the unique advantage of the system featuring an ultrathin membrane is that it separates the abiotic environment from the incompatible microbial environment on the nanoscale, thereby minimizing resistance losses and enabling scalability.

f) In table 1, several numbers “jumped” and should be fixed.

R: We have corrected this error.

Overall, the suggested system is interesting, but not novel enough to be published in Nature communication. Also, suffers from some basic design problems that make it less attractive.

R: The reviewer misunderstood that the core goal is to simultaneously chemically separate, yet electrochemically couple, the microbial and inorganic catalysts on the nanoscale. The design of introducing a nanoscale silica membrane between microbial cells and oxide surface provides a first solution to this problem. Hence, the approach solves a major gap in existing bioelectrochemical systems used for chemical synthesis.

To clarify the goal of this work, the abstract, the first and third paragraphs of the main text, and the conclusions have been significantly revised. To clarify our experimental and membrane design, we have introduced an additional paragraph (lines 129-135) and additional sentences throughout the results and discussion presentation.

Reviewer #2:

I enjoyed very much this interesting manuscript that may prove to be revolutionary in the field of bioelectrochemical systems. For many years researchers have been developing BESs and needed to deal with current collection. The approach presented here avoids current collection issues for those systems where two reactions can be combined that do not require additional energy input (although I could think of some options where this could be dealt with as well, indirectly). For me, the key reason to do this is thus current collection, and not necessarily the occurrence of two incompatible reactions as this is often topical.

R: The concept we demonstrate here is designed to connect the oxidative and reductive reactions on the nanoscale and is targeted for BESs that do not need additional power input or output, i.e. that can operate in short circuit mode. However, the reviewer correctly points out that the nanoscale membrane introduced here could operate as a current collector in other bioelectrochemical systems that do not operate in short circuit mode, such as microbial fuel cells or microbial electrolysis cells. A persistent challenge in designing current collectors is the need to provide chemical separation between the current collector and the microbial catalyst to avoid biofouling and oxygen crossover (new Ref 44-46). As demonstrated in our work, these membranes would provide robust chemical separation and high conductivity needed for current collectors. Since a significant portion of the research into BESs is on MFCs, the use of our membrane as a current collector would be transformative to the field.

Additionally, the incompatibility of oxidative and reductive chemistry is frequently cited as a major challenge in bioelectrochemical systems that synthesize chemicals (Ref 4). For example, inorganic catalysts generate reactive oxygen species (Ref 15,19) or leach heavy metal ions (Ref 19,20) which kill the microbial catalysts, or microbial catalysts corrode inorganic catalysts (Ref 5) or generate undesired products, i.e. CH_4 in H_2 , via cross-reactions (Ref 3,6). As the reviewer implies, there are strategies to circumvent the incompatibility of microbial and inorganic catalysts in these bioelectrochemical systems, i.e. physical separation or use of a macroscopic membrane, but these introduce significant ohmic losses that impair scale-up (Ref 2). Therefore, a major challenge for these bioelectrochemical systems is chemically separating the microbial and inorganic catalyst without introducing ohmic losses (Ref 1,2,4). The results presented in this paper address this scientific gap.

In response to the reviewer's comments, we have significantly edited a paragraph in the results and discussion (lines 256-265) and provided a figure in the supporting information (Figure S1) that describes the use of these nanoscale membranes in a short-circuit geometry for chemical synthesis and as current collector for power production.

On the downside, the manuscript needs to be improved in terms of clarity. First, based on the abstract it is next to impossible to understand what is achieved.

R: The key achievement is a demonstration that microbial catalysts and inorganic catalysis can be chemically separated yet, electrochemically coupled on the nanoscale. Underlying this achievement is an nm-thick, proton conducting, gas blocking membrane with embedded wires that has the robustness and appropriate energetics to work with microbial catalysts.

In response to the reviewer's comment, the abstract has been rewritten for clarity, and specific data have been included (lines 32-35).

Why not call it a bioelectrochemical system, rather than biohybrid system which can be many things?

R: A variety of terms are used in the literature to describe a bioelectrochemical cell in which current flows between inorganic catalysts and microbial catalysts, including microbial electrolysis systems, microbial electrosynthesis systems, electrofermentation systems, artificial photosynthesis systems, and photosynthetic biohybrid systems.

We have clarified this terminology in the text (lines 38-42) and followed the reviewer's suggestion to use the more general term 'bioelectrochemical system'.

There is unfortunately no data in the abstract, and for the reader it is not clear that anodic oxidation of lactic acid is catalyzed by *Shewanella* leading to the reduction of an electron acceptor on the other side. I would also stress the key advantage of current collection here.

*R: Following the suggestion, data have been added to the abstract. The reviewer is correct that anodic oxidation of lactate by *Shewanella oneidensis* leads to the reduction at the cathode. We have clarified this in the text (lines 162-165).*

Second, the manuscript is quite elaborate on the construction of the multilayer system, and is subsequently quite brief on the effective experiments.

R: In response to the reviewer's comments, we have condensed our description of the characterization of the nanoscale membrane by shorting the text on p.6 and 7 of the original version by more than half, by transferring more detailed spectroscopic information to the SI. A new section was added to the SI that presents these details.

I would like to see electron balances for consumed lactate, showing consumption coupled to a charge transfer that can be quantified. New lactate is added during an experiment, however it is unclear whether the lactate was used up by then? Based on the low current I would not expect so.

R. Comparing the electron balance for consumed lactate to the current produced allows measurement of the Coulombic efficiency of the bioelectrochemical reactor, which is dependent on many factors including the gas-tightness of the reactor. Because of our reactor design, electron balances for lactate consumption are dominated by these other factors rather than the electron transfer route. Specifically, the steady-state current transfer to the electrode from the bacteria was in the μA range (Figure 3), corresponding to $10 \mu\text{M}$ changes in lactate concentration. This change is several orders of magnitude below the precision of lactate measurements by HPLC ($\sim 0.5 \text{ mM}$). Additionally, since the reactors were microaerobic rather than strictly anaerobic, we expected and observed significant lactate consumption in the absence of current flow due to low levels of aerobic respiration of lactate. Therefore, the electron balances do not give accurate information on the fidelity of the charge transfer.

However, the key question behind the reviewer's comment is whether all current collected at the Pt/SnO₂ anode flow through the outer membrane cyt c of *S. oneidensis* and through the molecular embedded wires – a central conclusion of our work. Two key control experiments in Figure 3B demonstrate that this is the molecular path for current flow: No significant current flows when embedded wires with a LUMO at $\sim -1.7\text{V}$ ('wrong wires') or when the cyt c that mediate electron transfer are not present in *S. oneidensis*. These control experiments demonstrate the electron flow we measure must occur through the cyt c and through the embedded wires.

In response to the reviewer's comments, we have clarified that our bioelectrochemical measurements were performed under microaerobic consumption (lines 218, 592) and that some lactate consumption occurred due to the presence of low levels of oxygen (lines 605-607).

And, was this experiment repeated?

R: Yes, this experiment was repeated 4 times (line 182).

Third, I miss an experiment where the Pt layer is directly used with an electron acceptor, e.g. ferricyanide or oxygen. Unless I am mistaken, I have the impression that in all cases an additional counter electrode (Pt wire) was introduced.

R: Yes, in all cases an additional counter electrode was introduced.

In response to the reviewer's comment, we have included a schematic of this in Figure S9.

Line 40: in one sentence too many combinations are made, which makes unclear what is possible. Additionally, "biohybrid systems" is a new term in a field that does not need new terms unless justifiable. Why not use an existing technology term?

R: We have simplified this sentence to clarify what is possible (line 39-42). As described in the response to general remarks above, we have also clarified our use of terminology.

**Line 46: centimeters are never needed in practice
Key is to discuss current collection here as well**

R: Centimeters, or a macroscopic ion exchange membrane, are frequently required in practice for chemical synthesis to avoid crossover of incompatible compounds between the microbial and inorganic catalysts. For example, Giddings et al. (Ref 22) carefully optimize the distance to cm.

By line 140ish I was wondering whether some of the verification outcomes should be discussed in supplementary information, enabling a stronger focus on the results obtained with *Shewanella*

R: In response to the reviewer's comments, we have condensed our description by shorting the text on p. 6 and 7 of the original version by more than half, by transferring more detailed spectroscopic information to the SI. A new section was added to the SI that presents these details.

Line 167: I wondered here how the Pt side of the system was connected to

R: A schematic has been added to the SI as a new Figure S9 that shows the electrochemical system.

Line 188 and further: can it be demonstrated that the electrode is not able to oxidize hydrogen? At low partial pressures, the lactic can be oxidized well by the *Shewanella* and the anode can work at quite low potentials still

R: *Shewanella oneidensis* can oxidize lactate and generate hydrogen under certain circumstances, so if hydrogen could be oxidized by the electrode, it might be an alternative means of generating the current we observe. However, if hydrogen indeed acted as a diffusive mediator, then we would expect both the 'no wires' anodes and the 'wrong wires' anodes to generate current. Since neither anodes produce current above the background, we can exclude the possibility that hydrogen is acting as a mediator.

In response to the reviewer's question, we have explained how the experiments shown in Fig 3B exclude the possibility that diffusible mediators reduce SnO₂ or Pt in the 'correct wires' anode.

Line 259: aeruginosa

R: The typo has been corrected

Figure 1: I realize the interest of putting all parts of the system at the same scale, however it would be more evident to the reader what is being achieved if instead of terminal complexes a full bacterium would be drawn here, oxidizing lactate and transferring electrons

R: In response to the comment, we have included a new figure (Figure S1) that shows the full oxidative and reductive reactions.

Reviewer: Korneel Rabaey

Reviewer #3:

This manuscript describes a study that utilizes nanoscale thin film formed by novel material, which was specifically designed to fit the electron transfer from Shewanella outer membrane cytochromes (75-78). The approach logically makes sense to me though the results did not indicate such (175-182). Compared to bare SiO₂ membrane, the embedding of PV3 wire enhanced the current density as well as the bacteria viability. However, compared to the bare electrode surface, Pt/SnO₂, the coated electrode surface is inferior in both current density and bacteria viability. If the bare Pt/SnO₂ electrode surface represented a positive control, the logic needs to be laid out more clearly. For example, in a lot of microbial fuel cell studies, the biocatalyst (the biofilms) was directly grown on the electrode surface without chemical or physical separation. Often time, the biggest ohmic loss was due to the introduction of a membrane in between anode and cathode. Therefore, the logic of present study does not come to me naturally. Why the ohmic loss would be reduced by introducing a membrane between the biocatalyst and anode? It would be beneficial if this part can be extrapolated more clearly.

Response: As described in the opening paragraph of the text, the work presents a new concept – a nanoscale membrane - to address the widely-recognized need to chemically separate the biotic and abiotic environments in bioelectrochemical systems without introducing large ohmic losses. The abiotic and biotic catalysts in bioelectrochemical systems require different chemical environments for optimal function. For example, inorganic catalysts generate reactive oxygen species (Ref 15,19) or leach heavy metal ions (Ref 19,20) which kill the microbial catalysts, or microbial catalysts corrode inorganic catalysts (Ref 5) or generate undesired products, i.e. CH₄ in H₂, via cross-reactions (Ref 3,6). Thus, these systems use physical distance or macroscale ion-exchange membranes to separate the oxidative and reductive reactions, and the associated ohmic resistances significantly decrease the energy efficiency and the possibility of scale up.

By developing a nanoscale membrane and placing it between the microbial cells and the inorganic catalyst, our work solves two problems simultaneously: (1) it brings the oxidative and reductive processes together on the shortest possible length scale to minimize ohmic losses and (2) it prevents degradation of the inorganic electrode by the microbes (e.g. fouling) and degradation of the microbial catalyst by the inorganic electrode (e.g. production of reactive-oxygen species). The bare electrode does not accomplish the second problem. Thus, our work demonstrates a nanoscale

membrane as the first such solution to these broadly acknowledged major challenges in bioelectrochemical systems.

In response to the reviewer's remarks, which are echoed by Reviewers #1 and 2, we have edited the the Abstract (lines 24-37), the first paragraph of the main text (lines 39-54), and the first sentence of the Conclusions (lines 271-273) to clarify the key innovations in our work.

Additionally, the results need to be better organized. There appeared to be more than 3 types of electrodes, Pt/SnO₂, SiO₂ on Pt/SnO₂, PV3-SiO₂ on Pt/SnO₂, and PV3-SiO₂-SO₃ on Pt/SnO₂, though the authors did not mention the last one. Overall, this approach appears to be promising. I would encourage authors to keep trying with different materials and put biocompatibility into one of the consideration.

R: To improve the presentation of the data and provide further clarity regarding the 4 types of constructs, we have added a new paragraph to clearly lay out the design of our experiments (lines 129-135 and 194-204).

24-25 needs to give a better description about such system, for example, many researchers use the term bioelectrochemical system (BES) or microbial electrochemical system (MES). Same for 40-42. Two recent published reviews that describe the connection between biocatalyst and abiotic electrode which might be helpful to discuss present topic. "Li, C., Lesnik, K.L., Liu, H., 2017. Stay connected: Electrical conductivity of microbial aggregates. *Biotechnology Advances* 35(6), 669-680." "Lovley, D.R., 2016. Happy together: microbial communities that hook up to swap electrons. *The ISME Journal* 11(2), 327-336."

R: A variety of terms are used in the literature to describe a bioelectrochemical cell in which current flows between inorganic catalysts and microbial catalysts, including microbial electrolysis systems, microbial electrosynthesis systems, electrofermentation systems, artificial photosynthesis systems, and photosynthetic biohybrid systems.

We have clarified this terminology in the text (lines 40-43) and followed the reviewer's suggestion to use the more general term 'bioelectrochemical system.'

The paper by Liu and coworkers directly articulates the challenge of direct extracellular electron transfer between microorganism and inorganic material under separation of the incompatible biotic and abiotic environments. This is achieved in our approach by a nanoscale membrane, thereby obviating the need of macroscopic separation of anodic and cathodic catalysis, and additionally providing a means for preventing degradation of inorganic surfaces by fouling.

In response to the reviewer's comment, both references have been added as new Refs. 11 and 12

26-28 44-46 & 58-60 Why the separation from millimeters to centimeters would be the central challenge? the reason is not clear enough. In a lot of BES study, the biocatalyst the biofilms was directly grown on the electrode surface without chemical or physical separation. It seems that the biggest ohmic loss is between the biomass and the abiotic materials, in this case the membrane. Introducing one more layer of membrane into the assembly seems to increase the overall ohmic resistance. The benefits of introducing a separation membrane will have to be discussed. Does the membrane have higher affinity with the biofilms of *Shewanella*? or help transfer electrons to electrode? or the separation can help exhaustion of catalyst, etc. The strategy to reduce the contact resistance will have to be discussed more thoroughly.

R: This important comment shows that reviewer misunderstood the core concept behind this work. As described in the responses to the general remarks above, the core concept is a new membrane simultaneously capable of 1) electrochemical coupling between oxidative and reductive processes on the nanoscale and 2) chemical separation of the incompatible microbial and inorganic catalysts. No such solution to these broadly acknowledged major challenges of the field of bioelectrochemical systems has been proposed before, and proof of concept is demonstrated here.

We have added new text to the results and discussion (lines 178-179) to emphasize that the direct attachment does not meet these needs.

58-60 In the results of the present study, the bare Pt/SnO₂ electrode seems to possess the higher current density and microbial growth than the SiO₂-PV3 coated Pt/SnO₂ electrode. It is hard to declare the incompatibility of the Pt/SnO₂ electrode.

R: The reviewer is correct that the bare electrode has higher current density and microbial growth. However, direct growth of biofilm on the inorganic surface does meet the second need to chemically separate the abiotic and microbial environments (Figure 3A).

To demonstrate this new concept, it was necessary to show that the membrane could electrochemically couple the SnO₂ and the bacteria, therefore the SnO₂ electrode was biased and connected to an external current to allow current measurements. However, this membrane is designed to enable chemical synthesis. If SnO₂ (or any other inorganic material) is used as a catalyst in a microbial electrolysis cell, e.g. O₂ to hydrogen peroxide reduction by depositing metal clusters on the SnO₂, the chemical or gas crossover in the absence of the membrane or physical separation would kill the bacteria. The concept presented here is a nanoscale membrane that permits the chemical environments on either side to be completely optimized for their respective function, without introducing ohmic losses.

In response, we have clarified the text as described in the response to the general remark above.

60-62 the linkage between the advance and the scalable device was not really cleared. How the advance can impact the scalability. The cathodic limitation is also a big part as well as the cost of material, and thus the device that could be benefited from this technique should be mentioned more clearly.

R: The advance described here eliminates the need to physically separate the abiotic and biotic catalysts, thus eliminating ohmic losses. For example, microbial electrosynthesis systems (Ref 10-12) also called biohybrid systems (Ref 15,19,20), typically separate a water-splitting catalyst on the anode from the bacterial catalyst on the cathode to avoid killing the bacteria via cross-over of reactive oxygen species (Ref 15,19) or leaching of toxic metals (Ref 19,20). This introduces ohmic losses on the order of ~250 mV, or ~25% of the total electrochemical cell potential. The advance here will reduce these ohmic losses to ~10 mV, making scale-up possible. A further critical consequence of nanoscale integration of the abiotic and biotic function is the immense design space (immense variety of nanostructure for assembling macroscale systems) it opens up for scale-up.

In response to the reviewer's comment, we have clarified the introduction (lines 39-54, 70-74) and added a new figure (Figure S1) to indicate how the nanoscale membrane would be used for scalable microbial electrolysis or microbial electrosynthesis systems.

75-78 This section is the key concept of the whole article. I would suggest the authors to discuss/justify this statement more clearly. Why the LUMO of the designed polymer would be

more close to the cytochrome? From the attachment examination, it seems that the polymer **has been designed according to the authors' intention but the performance indicates otherwise. Therefore, it is essential to further extrapolate the discussion.**

R: As shown in Figure S2 and described on p.5 of the text, the LUMO potential of the newly designed molecular wire is closer to the cytochrome potential by over 1 V compared to unfunctionalized wire molecule. The finding that the unfunctionalized wire gives no bacterial current (Figure 3B, red trace) while the -NO₂ and -CN functionalized wire shows bacterial current demonstrates that the newly designed wire performs as intended, contrary to the reviewer's comment.

The nanoscale membrane is not yet optimized, and next steps to accomplish optimization are described in the discussion (lines 229-236). In response to the comments, a sentence has been added (lines 237-238) to elaborate on specific chemical modifications towards this goal.

172-174 Is and Io should be placed closer. The current position is confusing.

R: This error has been corrected.

175-186 the results were confusing. And reactor containing the bare electrode produce more current than the reactor containing Certain membrane prevent the growth of bacteria. The biocompatibility of this SiO₂-PV3 thin film should be discussed.

R: While the bare electrode has higher current density and microbial growth, there is no chemical separation between the abiotic and microbial environments (Figure 3A). As described above, the point is that the new nanoscale membrane plays two critical functions, namely electrochemically coupling oxidizing and reducing reactions of the system at the nanoscale and chemically separating the bacterial cells from an abiotic catalyst, including its incompatible reducing side.

The SiO₂-PV3 thin film supports robust growth of *S. oneidensis* under aerobic conditions (Figure S8), demonstrating that it is biocompatible.

We have added new text to the results and discussion (lines 178-179) to emphasize that the direct attachment does not achieve chemical separation.

From the current density plot it seems that the *Shewanella* was still in the phase of attaching/biofilm forming and current density could still increase over time. From the CLSM figure, we can see that the surface of electrode was far from occupied. Longer experiment period should be considered.

R: We did let the *Shewanella* attach for longer time, and we did not observe additional increases in current. We believe the relatively low density of cells could be improved by improving the density and energetic of the embedded wires. Nonetheless, this work establishes a core proof-of-concept that provides a foundation for future improvements.

205-208: The difference would be hard to tell if both current densities were only background level. The current density should not be negative. This provides no further support to the conclusion in line 210-212

R: The green trace of Figure 3B clearly shows that the mutant does not result in observation of bacterial current while *Shewanella o.* gives rise to significant current. Similarly, the red trace of Figure 3B shows that no current is detected for the mismatched wire. The difference is unambiguous and the

conclusion that the presence of outer membrane cytochromes and proper (but not yet optimized) matching of the orbital energetics of the embedded wires is therefore compelling.

237-239 it is hard to tell the link when both current densities were at the background level

R: The difference of bacterial current for *Shewanella* with and without the key outer membrane cytochromes is unambiguous, contrary to the statement by the reviewer as can readily be seen from Figure 3B: the green trace shows no current while the black trace gives significant current.

244-246 Current data is not supportive enough for the claim of present article.

R: The data presented unambiguously demonstrate that bacterial current is transmitted across the silica membrane by virtue of embedded molecular wires. It is reasonable to assume that optimization of the parameters as described on p.11 will further improve the bacterial current.

264-268 The change of inorganic properties of the membrane did not seem to enhance the current production, when compare to bare SnO₂/Pt electrode. Therefore, the claim was not supported by the data.

R: Contrary to the reviewer's statement, change of the electronic properties of the molecular wires has already been proven to enable bacterial current, while before the change there was none (Figure 3B, comparison of red and black trace). The conclusion is clearly supported by the data presented.

271-272 The "incompatible environment" of biocatalyst needs to be explained.

R: As explained in the Introduction (p.3 and p.4), the incompatibility refers to the cathodic and anodic reaction environments. Even on the anodic side, there is a long-term incompatibility of the bacterial and inorganic surface that is solved by inserting the silica membrane between the bacteria and the SnO₂ surface.

633-634 Is that a question for the reviewer?

R: The erroneous sentence has been deleted.

636-637 Please explain the 45-70° and 5° steps more, or give a citation properly.

R: In response, an explanatory sentence and a reference (Ref. 5) has been added on p. 32.

The scale of figure 1 should be adjusted according to their actually scale.

R: When Figure 1 is presented as a scaled drawing, the key structural and energetic features are difficult to discern. Therefore, we present Figure 1 as a schematic, and not as a scaled drawing.

Figure 3, what is the background, what is the microbial cells? it seems that the bare electrode has the highest density.

R: In Figure 3C, black regions are the underlying surface and white regions show the fluorescence from the fluorescent bacteria. The bare electrode does have the highest density, but as mentioned above, this system lacks the chemical separation needed for bioelectrochemical systems that synthesize chemicals.

Reviewers' Comments:

Reviewer #1:

Remarks to the Author:

I read the comments carefully, but unfortunately, I was not convinced that the paper is suitable for publication in Nature Communication.

Frei and his colleagues claimed that the paper novelty was misunderstood by the reviewers. The major novelty lays on the ability to separate between electron transfer and chemical reactions in the biotic/abiotic interface. Many different methods have been used for selective current collection. For example, self-assembly monolayers with redox molecules, redox mediators designed for selective oxidation/reduction processes (inorganic complexes, usually with enzymes) while interfere molecules are thermodynamically limited to react with the electrode. Furthermore, MR1-Cyt C allow selective current collection directly without oxidizing lactate directly over the electrode, as had been shown repeatedly. Indeed, the configuration presented here is interesting and maybe could lead to a unique some advances, but at this time with limited results compare to controls, a major thermodynamic flaw in design, as well as several previously published papers with the same method, dramatically limits its importance and novelty. Therefore, unfortunately, I recommend publishing elsewhere.

Reviewer #2:

None

Reviewer #3:

Remarks to the Author:

This manuscript describes a study that utilize nanoscale thin film formed by novel material, which was specifically designed to fit the electron transfer from Shewanella outer membrane cytochromes. The design of the thin membrane is novel but the authors seems to have a trouble on explaining how this novel design integrated with the bioelectrochemical system works. Therefore, two out of three reviewers "misunderstood" the key concept of the present article was not just a coincident.

I would suggest that the author to give a clear definition of the "new generation of the BES" because the design of membrane in present study may not be as attractive to those BES that does not need separation between anode and anodic microbial catalyst, for example, microbial fuel cells and microbial electrolysis cells focus on hydrogen production.

In the present manuscript, four types of electrodes were constructed to compare and evaluate the effectiveness of such approach, Pt/SnO₂, Pt/SnO₂+SiO₂, Pt/SnO₂+SiO₂+PV3, and Pt/SnO₂+SiO₂+PV3_SO3. Compared to bare SiO₂ membrane and the wrong wire, the embedding of PV3 wire enhanced the current density as well as the bacteria viability. However, compared to the bare electrode surface, Pt/SnO₂, the coated electrode surface inferior in both current density and bacteria viability. With revision, the experimental design is much more clear now.

The design of the thin membrane was well tailored by choosing the SnO₂ and the molecular wires. And the approach of present manuscript is well illustrated in the revised version.

The assembly of the electrical membrane layer was novel with especially allowing the penetration of proton. Taken together, I believe that this manuscript makes a valuable contribution to the current development of BES and may extend to other biological/abiotic integrated systems as well. Therefore, the manuscript should be accepted with minor revision based on this version.

24. Most researchers would consider the microbial cells on the anode/cathode as the catalyst. Microbial cells can decrease the overpotential and regardless the mechanism behind should be considered as the catalyst. However, it appears that in present manuscript, the authors considered the both microbial cells and the abiotic electrode materials as catalyst and/or sometimes just the abiotic electrode materials. However, the electrode material parts are not often considered as the catalyst when considering the overall construction cost of the reactor.

I would suggest to further clarify the use of terminologies to avoid confusion. Give clear definitions about microbial catalyst and the abiotic catalyst. And be more specific on what types of BES.

39 please define "new generation of BES". Maybe the authors should consider to give a definition of the target system more specific to BES in reference 9, 10. And please give a specific definition about the new generation of BES, what makes them new? Design of membrane in present study may not be as attractive to those BES that does not need separation between anode and anodic microbial catalyst.

41 Most MECs still use catalyst on cathode but not anode, especially those MEC aiming for hydrogen production.

247 may be changed to a metric unit

252 probably should go along with the term of BES

Point-by-point response to Reviewer Comments

Reviewer #2

Reviewer #2 in comments to the editor said they would have liked to see more information on substrate balances, and a test without external second electrode as this would improve the manuscript. The reviewer did comment that the manuscript could be published without these so it is up to you wherever you take the reviewers advice.

Response: We agree the additional information on the substrate balances and test without an external second electrode would be valuable information. However, these experiments require development of a new bioelectrochemical system containing our nanoscale silica membrane (like those shown in Supplementary Figure 1), which is outside the scope of this manuscript and will be addressed in future work.

Reviewer #3 (Remarks to the Author):

This manuscript describes a study that utilize nanoscale thin film formed by novel material, which was specifically designed to fit the electron transfer from Shewanella outer membrane cytochromes. The design of the thin membrane is novel but the authors seems to have a trouble on explaining how this novel design integrated with the bioelectrochemical system works. Therefore, two out of three reviewers “misunderstood” the key concept of the present article was not just a coincident.

I would suggest that the author to give a clear definition of the “new generation of the BES” because the design of membrane in present study may not be as attractive to those BES that does not need separation between anode and anodic microbial catalyst, for example, microbial fuel cells and microbial electrolysis cells focus on hydrogen production.

Response:

Per the reviewer’s request, we have clarified that the new generation of bioelectrochemical systems focuses on synthesis of chemicals rather than production of electricity or hydrogen (lines 40-1).

In the present manuscript, four types of electrodes were constructed to compare and evaluate the effectiveness of such approach, Pt/SnO₂, Pt/SnO₂+SiO₂, Pt/SnO₂+SiO₂+PV3, and Pt/SnO₂+SiO₂+PV3_SO3. Compared to bare SiO₂ membrane and the wrong wire, the embedding of PV3 wire enhanced the current density as well as the bacteria viability. However, compared to the bare electrode surface, Pt/SnO₂, the coated electrode surface inferior in both current density and bacteria viability. With revision, the experimental design is much more clear now. The design of the thin membrane was well tailored by choosing the SnO₂ and the molecular wires. And the approach of present manuscript is well illustrated in the revised version. The assembly of the electrical membrane layer was novel with especially allowing the penetration of proton. Taken together, I believe that this manuscript makes a valuable contribution to the current development of BES and may extend to other biological/abiotic integrated systems as well. Therefore, the manuscript should be accepted with minor revision based on this version.

24. Most researchers would consider the microbial cells on the anode/cathode as the catalyst. Microbial cells can decrease the overpotential and regardless the mechanism behind should be considered as the catalyst. However, it appears that in present manuscript, the authors considered the both microbial cells and the abiotic electrode materials as catalyst and/or sometimes just the abiotic electrode materials. However, the electrode material parts are not often considered as the catalyst when considering the overall construction cost of the reactor.

I would suggest to further clarify the use of terminologies to avoid confusion. Give clear definitions about microbial catalyst and the abiotic catalyst. And be more specific on what types of BES.

Response: We have clarified that the bioelectrochemical systems of primary interest are microbial electrolysis cells and microbial electrosynthesis systems (lines 24-25). Additionally, we have clarified that the microbial cells serve as oxidative catalysts and the abiotic materials serve as reductive catalysts in microbial electrolysis cells, and that these roles are switched in microbial electrosynthesis, artificial photosynthesis, and biohybrid systems (Lines 45-50).

39 please define “new generation of BES”. May be the authors should consider to give a definition of the target system more specific to BES in reference 9, 10. And please give a specific definition about the new generation of BES, what makes them new? Design of membrane in present study may not be as attractive to those BES that does not need separation between anode and anodic microbial catalyst.

Response: Per the reviewer’s request, we have clarified that the new generation of bioelectrochemical systems focuses on synthesis of chemicals rather than production of electricity or hydrogen (lines 40-1).

41 Most MECs still use catalyst on cathode but not anode, especially those MEC aiming for hydrogen production.

Response: We have clarified our language to indicate that any abiotic material that facilitates the electrochemical reactions, including carbon-based materials or noble metals, is consisted an abiotic catalyst (Lines 46-47).

247 may be changed to a metric unit

Response: We have changed the units from square inches to square cm (Line 252).

252 probably should go along with the term of BES

Response: We have made this correction (Line 256-7).